# The Relationship Between a Sustainable Healthy Lifestyle and Depression, Stress, and Anxiety: A Structural Model on the Mediating Role of Physical Literacy

**DOI:** 10.3390/healthcare13141646

**Published:** 2025-07-08

**Authors:** Mehmet Akarsu, İsmail İlbak, Zeliha Çavuşoğlu, Ratko Pavlović, Ana Maria Vulpe, Adina Camelia Șlicaru, Nicolae Lucian Voinea, Cristina Ioana Alexe

**Affiliations:** 1Department of Physical Education and Sports, İnönü University, 44050 Malatya, Türkiye; mehmet_akarsu@inonu.edu.tr; 2Institute of Health Sciences, İnönü University, 44050 Malatya, Türkiye; isma_ilbak@hotmail.com; 3Faculty of Sports Science, İnönü University, 44050 Malatya, Türkiye; zelihacavusoglu24@gmail.com; 4Faculty of Physical Education and Sport, University of East Sarajevo, 71123 East Sarajevo, Bosnia and Herzegovina; pavlovicratko@yahoo.com; 5Department of Physical Education and Sports Performance, “Vasile Alecsandri” University of Bacău, 600115 Bacău, Romania; lucian.voinea@ub.ro; 6Department of Physical and Occupational Therapy, “Vasile Alecsandri” University of Bacău, 600115 Bacău, Romania; slicaruadinacamelia@ub.ro

**Keywords:** psychological resilience, lifestyle behaviors, body awareness, health psychology, self-regulation

## Abstract

**Background:** It is well established that healthy lifestyle behaviors have significant effects not only on physical health but also on psychological well-being; however, the underlying mechanisms of these effects have yet to be fully elucidated. In this context, the aim of this study is to examine the relationships between a sustainable healthy lifestyle and levels of depression, stress, and anxiety, and to test the mediating role of physical literacy in these associations. **Methods:** This cross-sectional study was conducted among university students in Malatya, Türkiye, and a total of 652 voluntary participants were included. In the theoretical model of the study, a sustainable healthy lifestyle was positioned as the independent variable, depression, stress, and anxiety as dependent variables, and physical literacy as the mediating variable. Data were analyzed using structural equation modeling with bootstrapping to assess mediation effects. **Results:** Results obtained from the structural equation modeling indicate that a sustainable healthy lifestyle has significant and protective relationships with psychological well-being. Negative and significant relationships were identified between a sustainable healthy lifestyle and levels of depression, stress, and anxiety. Furthermore, higher levels of physical literacy were associated with lower levels of these psychological symptoms, with physical literacy playing a strong mediating role in these relationships. The model results revealed that lifestyle components such as regular physical activity, balanced nutrition, and sufficient sleep enhance individuals’ physical competence and awareness. Physical literacy was also found to have a significant negative relationship with depression, stress, and anxiety. **Conclusions:** These results indicate that physical literacy is a key variable not only for physical functioning but also for psychological resilience and well-being. Moreover, the impact of a sustainable healthy lifestyle on psychological symptoms appears to be largely mediated through physical literacy.

## 1. Introduction

In today’s rapidly changing living conditions, there has been an increase in environmental, social, and individual risk factors that threaten individuals’ psychological well-being and mental health [1,2]. This situation has led to a global rise in the prevalence of common mental disorders such as depression, stress, and anxiety, significantly impacting individuals’ quality of life [3,4,5,6]. Recent research has shown that individuals’ lifestyle choices are closely associated with these psychological problems [7,8]. For example, unhealthy lifestyle habits in individuals with polycystic ovary syndrome have been found to be related to depressive symptoms [9]. Similarly, high levels of physical activity among medical students have been reported to reduce the risk of depression [10]. These various examples indicate that lifestyle components strongly interact with mental health not only in healthy individuals but also in diverse risk groups.

The fundamental components of a healthy lifestyle include regular physical activity [11,12,13], balanced nutrition [14,15], adequate and quality sleep [16], and effective stress management [11,17]. The integrated implementation of these elements has been shown to reduce levels of depression, anxiety, and stress while supporting individuals’ psychological well-being [18,19]. In particular, physical activity not only directly alleviates symptoms of depression and anxiety but also indirectly enhances stress coping skills and strengthens social support systems [5,20,21,22]. The combined application of balanced nutrition and physical activity has been reported to yield more effective outcomes than isolated interventions [23]. Furthermore, quality sleep habits and mindfulness-based relaxation techniques also play a vital role in the maintenance of mental health [19,24]. A sustainable healthy lifestyle can be broadly defined as the consistent and long-term adoption of health-promoting behaviors—such as physical activity [25,26], nutrition [25,27]), sleep [27,28], and stress management [25,27] (—in a way that is adaptable to life changes and supports overall well-being. However, the sustainability of these behaviors depends on individuals possessing the necessary knowledge, skills, and motivational infrastructure to support their lifestyle choices. At this point, the concept of physical literacy emerges as a significant variable.

Physical literacy is a multidimensional competence domain that encompasses the knowledge, skills, motivation, and confidence necessary to support lifelong participation in physical activity [29,30,31,32]. The literature indicates that individuals with high levels of physical literacy tend to experience lower levels of depression, anxiety, and stress [33,34]. In this context, physical literacy is considered a mediating variable through which individuals can promote their mental health via physical activity [33]. Studies conducted on university students have shown that levels of physical literacy and mindfulness partially mediate the effects of psychological distress on life satisfaction [35,36]. Given that both physical literacy and sustainable healthy lifestyles emphasize self-directed and consistent health-promoting behaviors [37,38], it is plausible that physical literacy contributes to the maintenance of a sustainable healthy lifestyle by enhancing individuals’ ability and motivation to engage in such behaviors over time [39,40]. However, the nature of this relationship has not been sufficiently addressed in the literature.

Nonetheless, empirical evidence explaining the potential mediating role of physical literacy in the relationship between a sustainable healthy lifestyle and depression, stress, and anxiety remains limited. Therefore, there is a need to comprehensively investigate the effects of physical literacy on mental health and to clarify the underlying mechanisms of this relationship. In this context, the first aim of the present study is to examine the relationships of a sustainable healthy lifestyle to depression, stress, and anxiety; the second aim is to test the mediating role of physical literacy in this relationship.

## 2. Materials and Methods

### 2.1. Study Design

A cross-sectional digital survey was developed using the Google Forms platform (Google LLC, Mountain View, CA, USA) and was distributed to a broad sample of participants. The study is reported in accordance with the Checklist for Reporting Results of Internet E-Surveys (CHERRIES) [41] and the Strengthening the Reporting of Observational Studies in Epidemiology (STROBE) guidelines [42]. Ethical approval for the study was obtained from the Research and Publication Ethics Committee of Social and Human Sciences at Inönü University (Approval number: 16 on 5 February 2025).

The research was conducted in Malatya, Türkiye, specifically among undergraduate students at Inönü University. The study was carried out in an educational setting, where participants were recruited via institutional communication channels, including faculty mailing lists and student networks.

### 2.2. Participants

In the theoretical model of the study, a sustainable healthy lifestyle was positioned as the independent variable; anxiety, stress, and depression were defined as dependent variables; and physical literacy was considered a mediating variable. The minimum number of participants required for this study was calculated using G*Power software (version 3.1.9.7; University of Düsseldorf, Düsseldorf, Germany). Assuming an effect size of 0.02, an alpha level (α) of 0.05, a power (1-β) of 0.80, and five predictors, the analysis indicated that at least 647 participants were needed. Accordingly, a total of 652 participants who volunteered were included in the study. The research was conducted in accordance with the Declaration of Helsinki and relevant ethical guidelines. The demographic characteristics of the participants are presented in Table 1. All participants were students currently enrolled at Inönü University, and the study took place within a non-clinical academic environment.

The inclusion criteria for this study were as follows: participants had to be currently enrolled university students who voluntarily agreed to participate after being informed about the study’s purpose and procedures. Individuals were required to have access to the internet to complete the online survey administered via Google Forms and be capable of providing informed consent. Only those who completed all sections of the survey—ensured by mandatory response settings—were included in the analysis. The exclusion criteria involved any individuals who did not consent to participate, submitted incomplete responses, or attempted to participate more than once, as only single entries per participant were permitted. Additionally, individuals who were not part of the intended student population or whose academic affiliation could not be verified were excluded. While participants experiencing psychological distress were not explicitly excluded, the survey design included a debriefing with information about support services, acknowledging that the study was not intended to target individuals undergoing acute psychological crises.

### 2.3. Data Collection Tools

The order of the data collection tools used in the study was structured to minimize participants’ cognitive and emotional burden. First, a Demographic Information Form was administered to obtain basic demographic details such as age, gender, and physical activity status. Following this, the Healthy and Sustainable Lifestyle Scale was applied, which included more neutral questions regarding participants’ general lifestyle and behaviors. Third, the Perceived Physical Literacy Scale was used to assess participants’ perceptions of physical activity. Finally, the Depression, Anxiety, and Stress Scale (DASS-21), containing more sensitive content related to psychological conditions, was administered. This sequence was designed to prevent emotional bias from affecting responses to other scales and to enhance data reliability. Additionally, after completing the DASS-21, participants were informed about available support services if needed.

### 2.4. Healthy and Sustainable Lifestyle Scale

To assess individuals’ levels of sustainable healthy lifestyle, the Healthy and Sustainable Lifestyle Scale developed by Choi & Feinberg [43] and adapted into Turkish by Gökkaya [44] was used. This scale consists of 28 items rated on a 5-point Likert scale (1 = Strongly Disagree to 5 = Strongly Agree). Confirmatory factor analysis showed a good model fit, and the Cronbach’s alpha internal consistency coefficient was reported as high (α > 0.80) [44]. For the current study sample, the confirmatory factor analysis yielded acceptable fit indices (χ^2^ = 142.720, Df = 18, CFI = 0.985, TLI = 0.973, NFI = 0.979, IFI = 0.985, RMSEA = 0.064, SRMR = 0.027), and the Cronbach’s alpha for this study was calculated as 0.883.

### 2.5. Perceived Physical Literacy Scale

To evaluate the level of physical literacy, the Perceived Physical Literacy Scale developed by Sum et al. [45] and adapted into Turkish by Yılmaz & Kabak [46] was utilized. The scale includes 9 items and is rated on a 5-point Likert scale (1 = Strongly Disagree to 5 = Strongly Agree). Confirmatory factor analysis indicated good fit indices and high reliability [46]. In this study, the confirmatory factor analysis results for the scale showed acceptable model fit (χ^2^ = 87.536, Df = 22, CFI = 0.971, TLI = 0.952, NFI = 0.961, IFI = 0.971, RMSEA = 0.068, SRMR = 0.033), and the Cronbach’s alpha coefficient was calculated as 0.849.

### 2.6. Depression, Anxiety, Stress Scale-21 (DASS-21)

To assess levels of depression, anxiety, and stress, the DASS-21 developed by Lovibond & Lovibond [47] and adapted into Turkish by Sariçam [48] was used. The scale consists of 21 items rated on a 4-point Likert scale ranging from 0 = Never to 3 = Almost Always, based on the participants’ experiences over the past week. The Turkish version demonstrated good model fit and high internal consistency with reported Cronbach’s alpha coefficients of α = 0.87 for depression, α = 0.85 for anxiety, and α = 0.81 for stress. For the present sample, confirmatory factor analysis showed acceptable fit indices (χ^2^ = 453,596, Df = 97, CFI = 0.921, TLI = 0.902, NFI = 0.902, IFI = 0.921, RMSEA = 0.075, SRMR = 0.042), and the Cronbach’s alpha values were α = 0.813 for depression, α = 0.790 for anxiety, and α = 0.815 for stress.

### 2.7. Data Collection Process

The data collection process began in April 2025, following ethical approval from the Research and Publication Ethics Committee of Social and Human Sciences at Inönü University. Participants were informed about the purpose and scope of the study, and data were collected based on voluntary participation. The survey link was shared with university students via social media groups, emails, and student communities. Informed consent was obtained before participation, and participants were assured that their involvement was entirely voluntary and that the data would be used solely for scientific purposes. The survey took approximately six minutes to complete, and each participant was allowed to participate only once. After collecting demographic information, the scales were administered in the predetermined order. To prevent missing data, each question was marked as mandatory, ensuring full completion. All responses were collected anonymously, and the confidentiality of personal information was strictly maintained. The data were securely stored in digital environments and used solely for the purposes of this research.

### 2.8. Data Analysis

Data analysis was conducted using the JASP statistical software (version 0.18.3.0; University of Amsterdam, Amsterdam, Netherlands). The normality of the data distribution was assessed by examining skewness and kurtosis values within the ±2 range [49,50,51]. The results indicated that the data were normally distributed (Table 2). Consequently, Pearson correlation analysis was employed to examine the relationships among the variables. Given the high correlations among the psychological outcome variables (anxiety, stress, and depression), multicollinearity was assessed via tolerance and variance inflation factor (VIF) values, all of which remained within acceptable limits (VIF < 5, Tolerance > 0.20) [52]. In the study, a sustainable healthy lifestyle was treated as the independent variable, while anxiety, stress, and depression were the dependent variables. Physical literacy was considered a mediating variable in explaining this relationship.

To test the hypothesized mediation model, structural equation modeling (SEM) was conducted using JASP’s SEM module. The model fit was evaluated using conventional indices, including the Chi-square/df ratio (χ^2^/df), the Comparative Fit Index (CFI), the Tucker–Lewis Index (TLI), the Root Mean Square Error of Approximation (RMSEA), and the Standardized Root Mean Square Residual (SRMR). Acceptable model fit was determined based on widely used criteria (e.g., CFI and TLI > 0.90, RMSEA and SRMR < 0.08).

To test the statistical significance of the mediation effect, a bootstrapping method with 5000 samples was applied. The 95% confidence intervals obtained from this procedure were used to evaluate the significance of the indirect effects; the absence of zero within these intervals indicated statistically significant indirect effects.

## 3. Results

The findings obtained within the scope of the research are presented below through tables and figures. In this context, the relationships between a healthy and sustainable lifestyle, physical literacy, anxiety, stress, and depression are presented in Table 2.

According to the correlation results presented in Table 2, a significant negative relationship was found between a healthy and sustainable lifestyle and anxiety (r = –0.479, *p* < 0.001). Similarly, a significant negative relationship was observed between a healthy and sustainable lifestyle and stress (r = –0.446, *p* < 0.001) and between a healthy and sustainable lifestyle and depression (r = –0.477, *p* < 0.001). Moreover, there was a significant positive relationship between a healthy and sustainable lifestyle and physical literacy (r = 0.497, *p* < 0.001). Physical literacy was found to be significantly negatively related to anxiety (r = –0.760, p < 0.001), stress (r = –0.703, *p* < 0.001), and depression (r = –0.689, *p* < 0.001). Additionally, significant positive correlations were observed between anxiety and stress (r = 0.666, *p* < 0.001), anxiety and depression (r = 0.712, *p* < 0.001), and stress and depression (r = 0.680, *p* < 0.001).

Table 3 presents the findings on the mediating role of physical literacy (PL) in the relationship between a healthy and sustainable lifestyle (HSL) and anxiety, stress, and depression. Regarding direct relationships, the relationship between PL and anxiety was found to be statistically significant (β = –0.694, z = –24.065, 95% CI [–0.750, –0.637]), as was the direct relationship between HSL and anxiety (β = –0.149, z = –4.648, 95% CI [–0.212, –0.086]). Similarly, the direct relationship between PL and stress was significant (β = –0.640, z = –20.193, 95% CI [–0.702, –0.577]), along with the direct relationship between HSL and stress (β = –0.143, z = –4.063, 95% CI [–0.212, –0.074]). The relationship between PL and depression was also statistically significant (β = –0.599, z = –18.751, 95% CI [–0.662, –0.537]), as was the relationship between HSL and depression (β = –0.200, z = –5.627, 95% CI [–0.270, –0.130]). Furthermore, HSL was found to significantly predict PL (β = 0.553, z = 14.612, 95% CI [0.479, 0.627]).

Regarding indirect relationships, the mediating role of PL in the relationship between HSL and anxiety was found to be significant (β = –0.384, z = –12.490, 95% CI [–0.444, –0.324]). Similarly, the indirect relationships between HSL and stress (β = –0.354, z = –11.838, 95% CI [–0.412, –0.295]) and between HSL and depression (β = –0.331, z = –11.526, 95% CI [–0.388, –0.275]) through PL were also significant. These results confirm the mediating role of PL in the relationship between HSL and psychological symptoms (anxiety, stress, and depression).

Regarding total relationships, HSL was found to have significant relationships with anxiety (β = –0.533, z = –13.919, 95% CI [–0.608, –0.458]), stress (β = –0.497, z = –12.736, 95% CI [–0.573, –0.420]), and depression (β = –0.532, z = –13.875, 95% CI [–0.607, –0.457]). The variance explained by HSL and PL was 59% for anxiety, 51% for stress, 50% for depression, and 25% for PL.

The structural equation model examining the mediating role of physical literacy in the relationship between a sustainable healthy lifestyle (HSL) and anxiety, stress, and depression is illustrated in Figure 1. As shown in the model, the direct relationship between HSL and physical literacy (PL) is significant and positive (β = 0.55). In contrast, the direct relationships between HSL and anxiety (β = –0.15), stress (β = –0.14), and depression (β = –0.20) are negative. Physical literacy shows strong negative relationships with anxiety (β = –0.69), stress (β = –0.64), and depression (β = –0.60). Additionally, the model indicates a strong mediating role of PL, suggesting that the relationship between HSL and psychological symptoms is largely mediated through PL. The model also shows that stress has positive effects on both anxiety (β = 0.12) and depression (β = 0.18).

The overall fit of the structural equation model was evaluated using conventional goodness-of-fit indices. The results indicated an excellent model fit: χ^2^ = 15.540, df = 5, CFI = 0.994, TLI = 0.989, NFI = 0.992, IFI = 0.994, RMSEA = 0.057, and SRMR = 0.011. These values demonstrate that the proposed mediation model exhibits a very strong and excellent fit to the observed data.

## 4. Discussion

This study aimed to examine the relationship between a sustainable healthy lifestyle and symptoms of depression, stress, and anxiety, and to determine the mediating role of physical literacy within this relationship. The findings revealed that maintaining a healthy and sustainable lifestyle significantly contributes to psychological well-being. Specifically, significant and negative relationships were identified between a sustainable healthy lifestyle and the levels of depression, stress, and anxiety. Furthermore, as levels of physical literacy increased, these psychological symptoms decreased; physical literacy was found to play a significant and strong mediating role in these relationships. These findings suggest that healthy lifestyle behaviors provide not only physiological but also substantial psychological protective benefits.

According to the structural equation model, the relationship of a healthy, sustainable lifestyle to physical literacy was significant and positive. This indicates that lifestyle components such as regular physical activity, healthy eating, and adequate sleep contribute to enhancing individuals’ physical competence and awareness. Additionally, physical literacy was found to have significant negative relationships with depression, stress, and anxiety. This underscores that physical literacy is not only a determinant of physical functionality but also a key variable in psychological well-being. The study also revealed that the effects of a healthy, sustainable lifestyle on psychological symptoms are largely mediated through physical literacy. Thus, physical literacy may be considered a mechanism through which the psychological effects of healthy lifestyle behaviors are channeled and guided. Thus, physical literacy may be considered a mechanism through which the psychological relationships of healthy lifestyle behaviors are channeled and guided. These findings resonate with previous studies highlighting that emotional intelligence fosters university students’ happiness through the satisfaction of basic psychological needs, particularly competence and autonomy. While our model emphasizes physical literacy, it is likely that both constructs reflect broader self-regulatory mechanisms that facilitate psychological adaptation and resilience in academic contexts [53].

The findings are consistent with existing literature. Individuals with higher levels of physical literacy tend to exhibit better psychological health indicators. For instance, negative associations have been reported between physical literacy and internalization problems or overall psychological difficulties [54]. Moreover, physical literacy has been shown to play a crucial role in reducing burnout through resilience [55]. In children, higher levels of physical literacy have been associated with better quality of life [56]. In line with these findings, the current study also confirms physical literacy as a mediating mechanism that supports psychological health.

The correlation results obtained in this study further support these conclusions. Moderate negative relationships were found between a sustainable healthy lifestyle and depression, stress, and anxiety, while the correlations between physical literacy and these psychological symptoms were even stronger. The literature provides substantial evidence that lifestyle factors such as physical activity and nutrition significantly affect mental health. Physical activity has been shown to exert both direct and indirect effects in reducing symptoms of anxiety and depression [5]. A systematic review has highlighted the consistency of these effects across diverse demographic groups [57]. Physical literacy is also recognized as a significant factor for mental health. Individuals with adequate physical literacy tend to engage in higher levels of physical activity and report fewer psychological symptoms [33]. Furthermore, across various cultural contexts, individuals with higher healthy lifestyle scores have been found to experience lower levels of depression and anxiety [9,58]. These findings suggest that enhanced bodily awareness and movement competence contribute effectively to regulating emotions such as anxiety and internal restlessness.

On the other hand, the positive relationships of stress on both anxiety and depression observed in the model indicate that stress is a core trigger of psychological disorders. This finding aligns with Lazarus and Folkman’s [59] stress theory, which posits that when individuals lack adequate coping mechanisms, the risk of psychological maladjustment increases. In this study, increased levels of stress were accompanied by higher levels of anxiety and depression. These results support the preventive role of lifestyle behaviors and physical literacy in enhancing stress management skills.

The explained variances in the model are also noteworthy. Together, sustainable lifestyle and physical literacy accounted for 59% of the variance in anxiety, 51% in stress, and 50% in depression. These high levels of explained variance demonstrate that the model has strong explanatory power and that the findings are grounded in robust statistical evidence. Additionally, the fact that a sustainable lifestyle explained 25% of the variance in physical literacy highlights the significant role of lifestyle variables in the development of physical literacy.

Accordingly, the relationship between a sustainable lifestyle and psychological symptoms should not be viewed merely as a superficial correlation but rather as an explanatory and potentially causal association. The growing body of literature suggests that healthy behaviors such as regular physical activity and balanced nutrition positively impact individuals’ psychological resilience [60,61,62]. Similarly, physical literacy has been shown to contribute to individuals’ psychological well-being and reduce the severity of psychological symptoms [24,55,63,64,65]. The variance ratios obtained in this context are both theoretically and practically significant and emphasize the importance of interventions based on physical literacy in the development of psychological health.

Based on the practical implications of this study, the following recommendations can be made: For educators, health professionals, and policymakers, it is not sufficient to simply encourage individuals to adopt physical activity habits. It is equally important to enhance individuals’ levels of physical literacy—ensuring that their relationships with their bodies are conscious, competent, and sustainable. In this regard, integrating physical literacy-based approaches into schools, sports clubs, and community-based health programs can be effective in promoting both physical and psychological health. Focusing on this dimension in intervention programs aimed at adolescents and young adults can offer a strategic contribution to supporting mental health.

Despite the valuable insights provided by this study, several limitations should be acknowledged. First, the use of a cross-sectional design restricts the ability to infer causality among the variables. While the structural equation model reveals meaningful associations, only longitudinal or experimental designs can confirm directional relationships over time. Second, the reliance on self-report instruments may have introduced biases such as social desirability and common method variance. Although validated scales were used, objective or behavioral measures (e.g., wearable fitness trackers or observational assessments of physical literacy) could provide more robust data in future research. Additionally, the sample was limited to university students, which may constrain the generalizability of the findings to broader populations, including older adults or individuals with different educational or socioeconomic backgrounds. The use of a digital survey distributed via convenience sampling may also limit representativeness, as individuals without internet access or lower digital literacy could have been excluded. Lastly, cultural and contextual factors unique to the Turkish setting in which the study was conducted may influence lifestyle behaviors and perceptions of mental health and physical literacy. Future studies should consider cross-cultural comparisons to examine the universality of these findings.

## 5. Conclusions

In conclusion, this study has filled a significant gap in the field by revealing the holistic relationships of a healthy, sustainable lifestyle and physical literacy on psychological well-being. The findings demonstrate that it is not only the act of being physically active that contributes to mental health, but also the manner in which these activities are performed consciously, competently, and meaningfully. Physical literacy has clearly emerged as a fundamental component that bridges healthy lifestyle behaviors and psychological symptoms. In this context, if the goal is to prevent psychological disorders and enhance mental resilience, physical literacy should be considered a central component in educational, healthcare, and community-based programs. Future research employing longitudinal and multidimensional designs will further enrich the theoretical and practical knowledge base in this area by offering deeper insights into these relationships.

## Figures and Tables

**Figure 1 healthcare-13-01646-f001:**
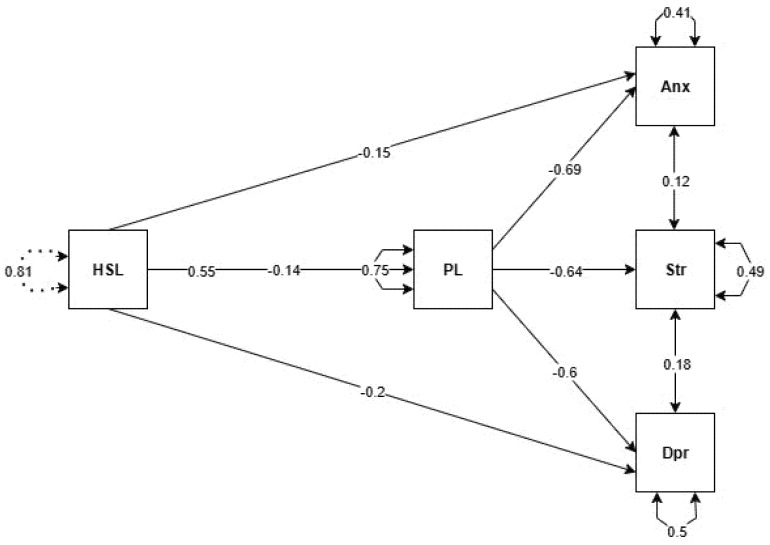
Model created for the mediating role of physical literacy in the relationship between a healthy and sustainable lifestyle and anxiety, stress, and depression. HSL  =  healthy and sustainable lifestyle, Anx  = anxiety , Str  =  stress, Dpr  = depression, PL = physical literacy.

**Table 1 healthcare-13-01646-t001:** Demographic characteristics of the participants.

Variable		$\bar{X}$	SD
Age (Years)		22.27	11.04
		n	%
Gender	Male	386	59.20
	Female	266	40.80
Engagement in Physical Activity	Yes	274	42.02
	No	378	57.98

**Table 2 healthcare-13-01646-t002:** Correlations among variables, descriptive statistics, and normality indicators.

Değişkenler	HSL	Anx	Str	Dep	PL	$\bar{X}$	SD	Skewness	Kurtosis
HSL	-	−0.479	−0.446	−0.477	0.497	3.478	0.898	−0.728	0.414
Anx	-	-	0.666	0.712	−0.760	1.213	0.851	0.861	0.777
Str	-	-	-	0.680	−0.703	1.309	0.802	0.656	0.475
Dep	-	-	-	-	−0.689	1.434	0.904	0.549	0.112
PL	-	-	-	-	-	3.703	0.786	−0.780	0.975

HSL  =  healthy and sustainable lifestyle, Anx  = anxiety , Str  = stress, Dep  = depression, PL = physical literacy, $\bar{X}$ = mean, SD = standard deviation.

**Table 3 healthcare-13-01646-t003:** Findings on direct, indirect and mediating relationships.

Direct Relationships								%95 Confidence Interval		
					β	Std. Error	z-Value	*p*	Lower	Upper
PL	→	Anx			−0.694	0.029	−24.065	<0.001	−0.750	−0.637
HSL	→	Anx			−0.149	0.032	−4.648	<0.001	−0.212	−0.086
PL	→	Str			−0.640	0.032	−20.193	<0.001	−0.702	−0.577
HSL	→	Str			−0.143	0.035	−4.063	<0.001	−0.212	−0.074
PL	→	Dep			−0.599	0.032	−18.751	<0.001	−0.662	−0.537
HSL	→	Dep			−0.200	0.036	−5.627	<0.001	−0.270	−0.130
HSL	→	PL			0.553	0.038	14.612	<0.001	0.479	0.627
**Indirect Relationships**								**%95 Confidence Interval**		
					**β**	**Std. Error**	**z-Value**	** *p* **	**Lower**	**Upper**
HSL	→	PL	→	Anx	−0.384	0.031	−12.490	<0.001	−0.444	−0.324
HSL	→	PL	→	Str	−0.354	0.030	−11.838	<0.001	−0.412	−0.295
HSL	→	PL	→	Dep	−0.331	0.029	−11.526	<0.001	−0.388	−0.275
**Total Relationships**								**%95 Confidence Interval**		
					**β**	**Std. Error**	**z-Value**	** *p* **	**Lower**	**Upper**
HSL	→	Anx			−0.533	0.038	−13.919	<0.001	−0.608	−0.458
HSL	→	Str			−0.497	0.039	−12.736	<0.001	−0.573	−0.420
HSL	→	Dep			−0.532	0.038	−13.875	<0.001	−0.607	−0.457

HSL  =  healthy and sustainable lifestyle, Anx  = anxiety, Str  =  stress, Dep  = depression, PL = physical literacy, → = direction of relationship.

## Data Availability

The original contributions presented in this study are included in the article. Further inquiries can be directed to the corresponding authors.

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
