# Peer review of "The Relationship Between a Sustainable Healthy Lifestyle and Depression, Stress, and Anxiety: A Structural Model on the Mediating Role of Physical Literacy"

_healthcare, 2025, doi:10.3390/healthcare13141646_

Round 1
Reviewer 1 Report
Comments and Suggestions for Authors
First of all, I would like to thank the editor for the opportunity to review this paper.
From the perspective of the subject matter addressed, I believe that the authors have made a valuable contribution to the field, enhancing our understanding of the analyzed topic.
I think the article is actually very good, with an original idea, methods and measures well described and a very well-designed study.
I just have one suggestion for the introduction.
I think the introduction could be improved. In my opinion, while the concept of physical literacy is sufficiently described, it could be useful to better define the construct of Sustainable Healthy Lifestyle. Furthermore, the introduction would be enhanced with a description of the links between Sustainable Healthy Lifestyle and Physical Literacy (that is very interesting in your study, but it is not directly explained).
Author Response
Dear Reviewer 1
Time is more than precious. Thank you for taking the time to review our manuscript. Your advice was appreciated. Attached are our responses.
Thank you

Reviewer 2 Report
Comments and Suggestions for Authors
The manuscript examines the relationship between sustainable healthy lifestyle behaviors and psychological symptoms (depression, stress, and anxiety), with physical literacy as a mediating variable. The topic is relevant, especially in the context of student well-being and health promotion. The rationale for the study is clearly presented, the use of structural equation modeling is appropriate, and the instruments are validated and adapted for the population under study.
The authors are commended for selecting a sufficiently large and statistically powered sample, and for reporting the main analyses with clarity. The structure of the paper is generally sound, and the discussion draws meaningful connections to previous findings in the field. There are, however, a few areas that would benefit from revision.
First, throughout the manuscript, the p-values should be written in italics (p < .001), following standard formatting conventions. Leading zeros in p-values should also be removed (e.g., p < .05, not p < 0.05).
Second, while the individual measurement scales are supported by appropriate model fit indices, the manuscript does not report overall model fit values for the structural equation model. This would allow readers to better evaluate the robustness of the proposed model.
Another point concerns the high correlations reported between the three psychological outcome variables (anxiety, stress, and depression), all exceeding r = .66. While these are expected to be related, it would be useful to mention whether multicollinearity was assessed, and if so, how it was addressed in the model.
One additional suggestion refers to the Discussion section, which would benefit from a deeper theoretical elaboration regarding the mechanisms through which physical literacy may promote psychological well-being. The authors interpret physical literacy primarily as a mediating behavioral construct; however, it may also be theoretically related to constructs such as self-regulation, competence, or psychological needs satisfaction, as conceptualized within self-determination theory. To strengthen this point, the authors may consider integrating a brief reference to the following study, which supports this view (https://doi.org/10.3390/psycholint6040055). This article shows how emotional intelligence promotes well-being through competence and autonomy, and may offer a valuable parallel mechanism that reinforces the authors’ argument about physical literacy as a gateway to psychological health. A suggested insertion point would be in the Discussion section, following the statement: “Thus, physical literacy may be considered a mechanism through which the psychological effects of healthy lifestyle behaviors are channeled and guided.” The addition of the following sentence would be appropriate: "These findings resonate with previous studies highlighting that emotional intelligence fosters university students’ happiness through the satisfaction of basic psychological needs, particularly competence and autonomy. While our model emphasizes physical literacy, it is likely that both constructs reflect broader self-regulatory mechanisms that facilitate psychological adaptation and resilience in academic contexts."
The paper offers useful insights and presents its findings with clarity. The revisions suggested here are modest in scope and aim to improve the precision and completeness of the manuscript before publication.
Author Response
Dear Reviewer 2
Time is more than precious. Thank you for taking the time to review our manuscript. Your advice was appreciated. Attached are our responses.
Thank you

Reviewer 3 Report
Comments and Suggestions for Authors
First, I would like to thank the authors for their work and the editor for the opportunity to revise the manuscript entitled The Relationship Between a Sustainable Healthy Lifestyle and Depression, Stress, and Anxiety: A Structural Model on the Mediating Role of Physical Literacy.
The subject is an important research subject. The manuscript aligns with the journal’s aims and scope. However, I have some concerns, and some points deserve modifications and suggestions for improving the manuscript's general overall appearance.
References
There are some old references in the manuscript. Are they still reliable, or are there newer references that can be used?
Abstract
In the abstract, I miss a description of where and in what context the study took place. The authors just mention that it was voluntary persons.
I also miss the study design statistical methods in the method part of the abstract.
In the result section of the abstract, the authors write direct negative effect. I do not think the word effect should be used here according to the cross-sectional design.
Introduction
In the introduction, I suggest that the authors may explain more about what they mean by a healthy lifestyle and refer to the new guidelines for different parts of a healthy lifestyle, such as the guidelines for physical activity from the World Health Organisation 2020.
Aim
In the aim the authors states In this context, the first aim of the present study is to examine the effects of a sustainable healthy lifestyle on depression, stress and anxiety.
In the Material and Method section, the authors write that the study is cross-sectional.
As they also note as a limitation in the discussion section cross-sectional studies are restricted for causal conclusions. The word effect indicates causal conclusions. I suggest using another word such as relationship or associations instead.
Material and Method
In the section about the participants, I miss a description of where the study was and in what context the study took place.
Under the data analysis section or in an own section I miss a deeper description of the statistical methods used. In the abstract result section, the results section and in the discussion, the authors write about structural equation model, and it needs a further explanation.
Results
In the results section, the authors write effect at least 12 times, including the text over Table 3. In the Material and Method section, the authors write that the study is cross-sectional.
As they also write as a limitation in the discussion part cross-sectional studies are restricted for causal conclusions. The word effect indicates causal conclusions. I suggest using another word such as relationship or associations, instead.
Discussion and conclusion
In the discussion, the word effect is used at least 6 times and once in the conclusions. It could mislead the reader to think that these are causal conclusion that could not be drawn from a cross-sectional study.
As they also write as a limitation in the discussion section cross-sectional studies are restricted for causal conclusions. This does not fit together with writing the word effect. The word effect indicates causal conclusions. I suggest using another word such as relationship or associations, instead.
Author Response
Dear Reviewer 3
Time is more than precious. Thank you for taking the time to review our manuscript. Your advice was appreciated. Attached are our responses.
Thank you

Round 2
Reviewer 3 Report
Comments and Suggestions for Authors
Thank you for allowing me to re-review the manuscript "The Relationship Between a Sustainable Healthy Lifestyle and Depression, Stress, and Anxiety: A Structural Model on the Mediating Role of Physical Literacy”
The authors have done a good job revising the manuscript, and I am satisfied with the response and revisions. The manuscript reads well, and the results are interesting.